

# Arctic Surface Snow Interactions with the Atmosphere: Spatio-Temporal Isotopic Variability During the MOSAiC Expedition

Moein Mellat[1,2], Amy R. Macfarlane[3,4] ★, Camilla F. Brunello[5] ★, Martin Werner[5], Martin Schneebeli[6], Ruzica Dadic[6], Stefanie Arndt[5], Kaisa-Riikka Mustonen[7], Jeffery M. Welker[7,8,9], Hanno Meyer[1]

[1] Alfred Wegener Institute Helmholtz Centre for Polar and Marine Research, 14401 Potsdam, Germany
[2] Institute for Environment Science and Geography, University of Potsdam, 14476 Potsdam, Germany
[3] UiT The Arctic University of Norway, Tromsø, Norway
[4] Northumbria University, Newcastle, United Kingdom
[5] Alfred Wegener Institute Helmholtz Centre for Polar and Marine Research, 27515 Bremerhaven, Germany
[6] WSL Institute for Snow and Avalanche Research SLF, Davos, Switzerland
[7] Ecology and Genetics Research Unit, University of Oulu, Oulu, Finland
[8] University of the Arctic (UArctic), Rovaniemi, Finland
[9] Department of Biological Sciences, University of Alaska Anchorage, Anchorage, AK, United States

*Correspondence to*: Moein Mellat (Moein.Mellat@awi.de)

★*These authors contributed equally to this study.*





**Abstract.**

The Arctic Ocean's snow cover is crucial in moderating interactions between sea-ice and the atmosphere, yet fully grasping its isotopic composition and the processes shaping it presents substantial challenges. This study employs a unique dataset from

the Multidisciplinary drifting Observatory for the Study of Arctic Climate (MOSAiC) expedition to explore the complex interactions between deposition processes and post-depositional changes affecting snow on Arctic sea ice. By examining 911 individual snow isotope measurements collected over a full year, we identify a clear layering within the snowpack: the top layer, with lower $\delta^{18}$O values and higher *d-excess* values, indicates fresh meteoric snowfall, while the bottom layer, affected by the sea ice beneath, shows higher $\delta^{18}$O values and lower d- *d-excess* values. By integrating these discrete snow samples

with continuous vapour isotope data, our research provides insight into interactions between snow and the atmosphere, as well as the processes that alter isotopic signatures within Arctic snow.

We observe a significant difference in $\delta^{18}$O values between snow and vapor during autumn, mainly due to delays in sampling after precipitation events, with *d-excess* ranges suggesting the impact of Atlantic moisture. Winter months exhibit sharp differences in $\delta^{18}$O and *d-excess* values, indicating kinetic fractionation amid extreme cold as the RV Polarstern traverses from

the Siberian to the Atlantic sector of the Arctic Ocean. Conversely, summer months display a convergence in isotopic signatures, reflecting conditions favouring equilibrium fractionation, highlighted by increased air temperatures and humidity levels. While $\delta^{18}$O in vapour readily responds to changes in air temperature and humidity, surface snow $\delta^{18}$O is influenced more by subsequent processes such as sublimation and wind-driven redistribution. Sublimation, intensified by the snow's prolonged surface residence and facilitated by the porosity of snow, plays a key role in isotopic enrichment. Wind-driven snow

redistribution, occurring 67% of the winter, led to a homogenised and depleted surface snow $\delta^{18}$O signal across the sea ice by spreading lower $\delta^{18}$O meteoric snow. This effect was especially pronounced in ridge snow profiles, where the top layers showed a uniform $\delta^{18}$O signal, in stark contrast to flat ice samples.

Furthermore, distinct isotopic patterns were detected along the MOSAiC expedition route from a region close to Samoylov Island to Fram Straight near Ny-Ålesund. Snow samples close to Samoylov Island exhibited notable seasonal $\delta^{18}$O variations,

which were indicative of a continental climate. In contrast, samples from Ny-Ålesund displayed more consistent fluctuations, influenced by steady Atlantic moisture.



## 1 Introduction

Changes in the Arctic's climate system are becoming more pronounced and are impacting the local ecosystems (Meng et al.,
2023). The region is undergoing an increase in temperature at a rate four times faster than the global average (Rantanen et al.,
2022), increases in precipitation and moistening of the atmosphere (Vihma et al., 2016; Kopec et al., 2016), and significant
loss of sea-ice (Jenkins and Dai, 2021; Smith et al., 2023; Sumata et al., 2023). These transformations in the Arctic system
also impact the Arctic's water cycle isotope geochemistry and the entirety of the hydrological cycle, which has the potential
to significantly affect the global climate system (Zhao et al., 2019; Yamanouchi and Takata, 2020). This is based in part on
more frequent extreme events such as warm air intrusions (Rinke et al., 2021; Bailey et al., 2021), shifting storm tracks (Akers
et al., 2020; Mellat et al., 2021; Aemisegger, 2018), warm air outbreaks (Leroy-Dos Santos et al., 2020; Rinke et al., 2021)
and greater degrees of evaporation from previously ice-covered seas (Boisvert et al., 2023). The consequences of Arctic system
changes have a wide-ranging impact on the complex interplay of sea-ice dynamics, climatic conditions, and hydrological
processes (Gottlieb and Mankin, 2024). Although the scientific community's understanding of the interaction between these
factors in the Arctic region is expanding (Shupe et al., 2022), our knowledge within the Arctic basin remains limited. To fully
document these changes and improve the robustness of future projections, it is important to quantitatively describe the Arctic
system based on local observations (Klein et al., 2024).

In the Arctic, snow on sea ice serves an essential role in the Earth's climate system, offering both high albedo properties that
aid in global cooling and acting as a superior insulator, thus playing a significant role in the thermal dynamics between the
ocean and the atmosphere (Webster et al., 2018; Goosse et al., 2018). Snow on sea ice affects the thermodynamic balance of
ice mass by providing insulation from atmospheric temperatures, especially during the darker and colder winter months
(Nicolaus et al., 2006; Perovich et al., 2017). This insulation significantly reduces the heat exchange between the ocean and
the atmosphere, influencing the growth and melting cycles of the sea ice. The interactions between snow, sea ice, and the
atmosphere involves feedback loops that regulate the Earth's climate at high latitudes (Webster et al., 2018; Feldl et al., 2020).
These interactions, however, are not fully understood due to the harsh and unstable conditions of the central Arctic. To address
this gap, we employ stable water isotope measurements to investigate the exchange processes between snow and atmosphere
that affect the isotopic signature of snow on Arctic sea ice.

Stable water isotopes ($\delta^{18}$O and $\delta^2$H) are powerful tracers for understanding the complex processes governing snow dynamics
in Arctic environments (Ala-Aho, 2021). Depositional processes, primarily involving moisture condensation due to low
temperatures and snowfall, define the initial isotopic signature of snow. However, post-depositional processes can result in
high spatial heterogeneity of snow accumulation across the sea-ice (Webster et al., 2018). These changes lead to a gradual
weakening of the correlation between snow isotopes and surface temperature, a phenomenon observed in the snow surface and
sub-surface layers (Mellat et al., 2024). This weakening highlights the complex interplay between snow and its environmental



conditions, demonstrating the significant impact of post-depositional processes such as sublimation, wind pumping, wind distribution, and snow metamorphism on the snow isotopic signature, complicating the interpretation of the original climatic signal (Casado et al., 2018; Laepple et al., 2018). The second-order parameter Deuterium excess (*d-excess*) helps to explore the post-depositional processes in surface snow (Sinclair and Marshall, 2008). Due to different sensitivity levels of $\delta^{18}O$ and

$\delta^2H$ to kinetic fractionation, *d-excess* is more prone to be influenced by the diffusion of water vapour associated with surface sublimation (i.e. a transition of snow directly into vapour without passing through a liquid phase) (Beria et al., 2018).

Snow metamorphism is a post-depositional process where snow grains experience structural changes to reach a state of thermodynamic stability (Colbeck, 1980; Vérin et al., 2022). This metamorphism occurs at the surface and within the snow

profile, largely driven by temperature gradients inducing vapour transport (Pinzer and Schneebeli, 2009). Snow metamorphism begins when snow-atmosphere interface is reduced, contributing to enhanced thermodynamic stability and triggering substantial shifts in the snow isotopic composition through vapour condensation and sublimation processes (Legagneux and Domine, 2005; Harris Stuart et al., 2023). During the Arctic melt season, two distinct but concurrent phenomena are observed: snow metamorphism occurring at the surface of the snowpack, and the formation of a surface scattering layer on the sea-ice in

contact with snow. Surface snow metamorphism impacts the albedo effect, where the highly reflective snow cover melts and exposes the less-reflective underlying sea-ice to solar radiation. The intense absorption of solar radiation leads to the formation of a porous, granular, and extremely fragile pillared structure atop the melting sea-ice, known as the surface scattering layer (Macfarlane et al., 2023b; Light et al., 2022). The presence of this layer facilitates the infiltration of freshwater into this layer through its pore spaces, thereby changing its isotopic composition.


Wind redistributes surface snow, leading to a differential accumulation on sea-ice, profoundly affecting the snow's physical and chemical characteristics (Nandan et al., 2023). The Arctic region experiences frequent and powerful wind events capable of transporting snow across distances ranging from several meters to multiple kilometres. Consequently, Arctic precipitation, initially deposited in a specific location, can mix with snow transported from proximal or distal origins, resulting in its

accumulation on sea-ice. This mixing effect alters the isotopic signature of the surface snow, challenging the interpretation of the isotopic composition solely as a direct indicator of local climate conditions (Hughes et al., 2021).

The vapour exchange between snow and the atmosphere plays an important role in gradually altering the original meteoric signal present in freshly fallen snow on Arctic sea-ice. Sublimation from snow into the atmosphere and deposition from the

atmosphere onto the snow is controlled by local atmospheric conditions and variations in surface heat fluxes. This exchange of vapour between the snow surface and the atmosphere alters the snow's composition through slow changes, significantly impacting the annual snow surface mass balance and the energy budget, as well as the isotopic composition (Gong et al., 2023).





Through the investigation of stable water isotopes of both discreet snow samples and continuous vapour measurements
(Brunello et al., 2023) collected during the Multidisciplinary drifting Observatory for the Study of Arctic Climate (MOSAiC)
expedition from September 2019 to October 2020 (Nicolaus et al., 2022), we obtained snow isotope dataset covering a full
year to observe processes between snow and the atmosphere. These isotopic signals aid in interpreting the alteration and
movement of water molecules between the snow and the atmosphere, while also offering insights into the physical and
chemical processes that impact the isotopic characteristics of surface snow. In this study, we examine the seasonality of
interaction between snow and atmosphere during MOSAiC, focusing on the isotopic compositions of the uppermost layer of
snow. We use vapour isotope measurements and data on precipitation events from continental stations to compare with the
isotopic composition of snow accumulating atop sea-ice, aiming to link the Central Arctic processes to the coastal sites. We
find that hydroclimatic variations in the Arctic significantly affect the snow cover's spatial and temporal alterations in height,
structure, and geochemical signature and we conclude that that meteoric signals on the snow surface are mostly overprinted
by the different post-depositional changes.

## 2 Methods

### 2.1 The MOSAiC expedition

The MOSAiC expedition consisted in a year-long drifting expedition across the Central Arctic. The German research vessel
Polarstern (PS), set sail from Tromsø, Norway, on September 20, 2019, ending its mission in Bremerhaven, Germany, on
October 12, 2020. This overall expedition was articulated  into five stages or 'legs' (Nicolaus et al., 2022).PS, anchored to a
selected ice floe, served as a dynamic platform for, among many other studies, continuous snow observations, particularly
during the initial three legs of the expedition (from October 2019 to mid-May 2020). After a logistical detour to Svalbard on
16 May 2020, the expedition's focus on snow research persisted when PS returned to the ice on 17 June 2020, and continued
summer measurements until the flow broke up on 31 July 2020. Then PS travelled to a location near the north pole (87 °N,
104 °E) to conduct further measurements in the central Arctic in August and September 2020.

### 2.2 Snow sample collection and laboratory analysis

#### 2.2.1 MOSAiC snow isotope and physical properties

During the MOSAiC expedition, a comprehensive snow survey was conducted, collecting 911 snow samples from various
profiles. These samples were organised into two distinct datasets. In dataset 1, snow samples were gathered from three specific
layers within the snow profiles: surface, middle, and bottom (Mellat et al., 2022). The surface layer was in direct contact with
the air, the middle layer was situated approximately halfway through the snow profile, and the bottom layer was directly above
the sea-ice surface. These samples, varying in depth due to the variable snow cover thickness, were collected using a density
cutter, stored in sampling bags, and maintained frozen until laboratory analysis. The collected samples were thawed at room





temperature prior to isotope analyses. Dataset 2 includes samples taken at every 3 cm of the snow profiles (Macfarlane et al.,
2022). Further, this dataset includes a vertical profiling approach, where snow samples, each quantifying 100 cm³, were
obtained and analysed for density on-site before being securely transported in sealed plastic cups. Following transport to PS,
these samples were melted for isotope measurements on the ship.

The samples from dataset 1 were processed and analysed at the ISOLAB facility of the Alfred Wegener Institute Helmholtz
Centre for Polar and Marine Research (AWI) in Potsdam, Germany. Here, the stable water isotopic composition, specifically
$\delta^{18}O$ and $\delta^2H$, was determined. Samples were thawed, transferred into 20 ml glass vials, sealed with parafilm tape, and stored
at 4 °C for subsequent isotopic analysis using mass spectrometers, employing equilibration techniques for high precision with
an accuracy of better than ±0.1 ‰ for $\delta^{18}O$ and ±0.8 ‰ for $\delta^2H$ (Meyer et al., 2000). Dataset 2's snow samples were analysed
in two different laboratories: the Swiss Federal Research Institute WSL and the ISOLAB Facility at the Alfred Wegener
Institute. At WSL, the samples' stable water isotopic composition was measured using a Los Gatos Research Isotopic Water
Analyzer. This process included multiple measurements per sample to guarantee measurement reliability and accuracy, with a
resulting measurement uncertainty of $\delta^{18}O \pm 1$ ‰ and $\delta^2H \pm 2$ ‰, and an accuracy of $\delta^{18}O \pm 0.5$ ‰ and $\delta^2H \pm 1$ ‰.

Determining the $\delta^{18}O$ and $\delta^2H$ involves calculating the isotopic ratio of heavy to light isotopes, compared to a standard, Vienna-
Standard Mean Ocean Water (V-SMOW), and presenting the difference in parts per thousand (per mil; ‰). In this study, we
use delta (δ) notation to represent the relative proportion of the sample's isotope ratio to that of the V-SMOW. Additionally,
the *d-excess*, a second-order parameter, is calculated using the equation $\delta^2H – 8 * \delta^{18}O$ (Craig, 1961).

To ensure the robustness of the data, a comparison was conducted between the two datasets. This involved measuring a subset
of 50 samples in both laboratories. By comparing the measurements of samples in both laboratories, we noticed a systematic
offset between dataset 1 and 2 independent of the laboratory measurements. A correction was required for dataset 2 samples
due to identified evaporative fractionation effects during storage. This correction was calculated by aligning the mean values
of the two datasets For the WSL dataset, the $\delta^{18}O$ was corrected by –6.4 ‰, and the $\delta^2H$ was corrected by –36.4 ‰ (Macfarlane
et al., 2022).

**2.2.2 Precipitation events from continental stations**

The Pan-Arctic Precipitation Isotope Network (PAPIN) was established in 2018 to improve our understanding of the Arctic
hydrological cycle using stable water isotope observations. The sample network currently includes more than 30 strategically
distributed stations spanning different Arctic climate zones, such as tundra, subarctic, maritime, and continental zones (Mellat
et al., 2021). In this study, we use stable water isotope data from event-based precipitation samples from two PAPIN stations
during the MOSAiC expedition. Samoylov Island (72 °N, 126 °E) and Ny-Ålesund (78 °N, 11 °W) stations were selected due
to their proximity to the track of MOSAiC and their data coverage. Samoylov Island is the closest continental station from the



start of the drift track in October 2019, and it is situated approximately 1450 km. Similarly, Ny-Ålesund, is located within a range of 200 to 550 km from PS in June and July 2020. Both stations provide spatial context for interpreting the MOSAiC observations.


Throughout MOSAiC, 90 precipitation events were collected at Samoylov Island over the complete year and 25 at Ny-Ålesund, spanning from May to September 2020. Each station was equipped with a precipitation gauge placed 0.5-1.5 m above the ground, and samples were collected immediately after each event. These were then stored in plastic screw-cap vials and refrigerated at 4 °C after sampling date, time, air temperature, and precipitation amounts were documented for each event. For

isotopic measurement, the samples were transferred to 2 ml septa-capped glass vials (for Ny-Ålesund), and 30 ml narrow-neck PE bottles (for Samoylov Island), and analysed for their $\delta^{18}$O and $\delta^2$H isotopes at the isotope labs at the University of Oulu, Finland (for Ny-Ålesund), and ISOLAB at AWI Potsdam (for Samoylov Island) with similar measurement precisions.

### 2.3 Complementary data

**Vapour isotope measurements during MOSAiC:** In addition to the snow isotope dataset, continuous, high-resolution

measurements of stable water isotopes of vapour were included into the analysis (Brunello et al., 2022a). These measurements were acquired through a Picarro L2140-i Cavity Ring-Down Spectroscopy (CRDS) instrument installed at around 29 m above the sea-ice level, which was operational throughout the entire campaign. This setup yielded a comprehensive dataset, including humidity, $\delta^{18}$O, and $\delta^2$H, with a temporal resolution of 1 Hz (Brunello et al., 2023).

**Meteorological data:** To assess the influence of meteorological conditions on the surface snow isotopes, we used the final, processed (Level 3) measurements and derived parameters, collected from an array of instruments located at the Met City site within the Central Observatory of MOSAiC (Cox et al., 2023). These data included air temperature (T) at 2 meters above the surface, relative humidity (RH) at 2 meters, latent heat flux (LHF) measured at 10 meters, and wind speed (WS) at 2 meters. Additionally, precipitation amounts along the trajectory of PS were extracted from the gridded European Centre for Medium-

range Weather Forecasts (ECMWF) fifth-generation reanalysis (ERA5) dataset (Hersbach et al., 2023).

**Long-term vapour and precipitation isotope data from continental stations:** To complement our dataset, we also analysed long-term simultaneous isotope data of vapour and precipitation from previous studies conducted at two continental stations, Samoylov Island and Ny-Ålesund. At the Lena River Delta in north-east Siberia, specifically at the research station on

Samoylov Island, vapour and precipitation data were collected (Bonne et al., 2020). A CRDS continuously recorded the near-surface water vapour isotopic composition from 2015 until 2017. Precipitation sampling was conducted after each rainfall and snowfall event. In Ny-Ålesund two Picarro CRDS were deployed (8 m above sea level), at the AWIPEV observatory building (Leroy-Dos Santos et al., 2020). A L1102-i Picarro instrument, was operational from May 2014 to May 2015, a L2130-i Picarro instrument replaced the first device and continued the measurements until the end of the campaign in September 2018. During



precipitation events, water or snow was sampled daily at midday. Over the 4.5 years, a total of 519 precipitation samples were
collected and analysed for their isotopes, offering a detailed long-term perspective on isotopic variations in the region.

## 3 Results

### 3.1 Snow profiles isotope variations throughout the MOSAiC expedition

This study investigates the seasonal variation of $\delta^{18}$O values in snow samples collected during the MOSAiC expedition from
October 2019 to September 2020. Figure 1 illustrates the relationship between snow height and isotope variations in the snow
on the sea-ice, relative to the median over the MOSAiC year (median $\delta^{18}$O$_{Snow}$: -15.60 ‰, *d-excess*: 5.1 ‰, as detailed in Table
S1). The figure reveals distinct seasonal shifts in snow height and snow isotopic composition compared to their respective
median values. From October to December 2019, $\delta^{18}$O anomalies (Figure 1a) are slightly above 0 ‰. Negative anomalies are
absent in this early phase. In January and February 2020, positive $\delta^{18}$O anomalies are pronounced, particularly in the lower
layers of the snow by up to 10 ‰, near the snow-sea-ice interface. Conversely, predominantly negative $\delta^{18}$O anomalies from
the median are observed in the upper layers of snow from January to May 2020, which may reach a difference from the median
of up to 25 ‰. June to August 2020 displays predominantly positive $\delta^{18}$O anomalies, which persists mostly in the lower layers
of snow, but the contrast top-to-bottom is less pronounced. At the same time, the snow height decreases significantly to values
below 30 cm from August onwards since no profiles were sampled from ridges anymore. Ridges, the elevated structures
formed on sea ice through the collision and compression of ice floes, serve as deposition zones where re-distributed snow
across the Arctic Ocean's sea-ice can gather at greater heights than on the surrounding flat surfaces. This accumulation process
enables the ridges to capture snow, up to 22% of all snow cover (Liston et al., 2018), that carries a distinct isotopic signature,
often more depleted in $\delta^{18}$O and indicative of meteoric origins. These trends (decreasing snow height and $\delta^{18}$O variability)
continue into the summer, with some $\delta^{18}$O anomalies up to 15 ‰ above the median. In September, the anomalies gradually
decrease but remain positive, similar to the conditions observed during the initial stage of the expedition. These are interrupted
by a phase of build-up and decay of the snow cover between January and August 2020, characterised by a higher $\delta^{18}$O
variability.

The *d-excess* anomalies (Figure 1b), generally confirm this subdivision: the early months (October to December 2019) are
characterised by generally negative deviations (down to -10 ‰) from the median across most samples. However, there are
positive anomalies (up to +10 ‰), especially at the bottom of the snow pack. The *d-excess* variability increases in January and
February 2020, with increasing snow height showing a large isotopic gradient between bottom and surface snow. During this
period, the lower layers of the snow cover near the sea-ice interface predominantly exhibit negative *d-excess* anomalies, while
the upper layers show positive deviations. In June and July 2020, the *d-excess* variability decreases, towards the surface snow
median. Notably, the period between January and April displays some of the highest positive anomalies recorded during the
entire expedition, exceeding 20 ‰ from the median in some instances. During the summer months, we observe lower *d-excess*



values and a less clear isotopic gradient. In September, the snow cover *d-excess* varies little around the median *d-excess* values, mirroring a stabilization in isotopic composition, that coincides with the transition from negative to positive temperatures, and freezing to melting conditions in the Arctic environment. Overall, the isotopic anomalies for both $\delta^{18}$O and

*d-excess* throughout the MOSAiC year exhibit a periodic pattern with similarities between early and late stages and a higher variability in between which are accompanied by changes in snow height.



**Figure 1 Snow profile samples collected during the MOSAiC expedition from October 2019 to September 2020. In 4 a), anomalies from the median δ¹⁸O (-15.6 ‰) are plotted against the snow height for each sample. Samples above 40 cm, typically represent snow accumulated on ridges, indicated by triangle symbols. Samples from snow heights below 40 cm are represented by circles. 4 b) shows the corresponding anomalies from median _d-excess_ (5.1 ‰) measurements across the snow height, with triangles to represent snow at the ridges and circles for flat ice areas. Grey shaded areas in the background of both panels represent periods when the PS was frozen in the Arctic Ocean's sea-ice, during which sampling was systematically conducted. White areas signify the intervals of transit between different segments of the expedition.**



## 3.2 Snow and vapour $\delta^{18}$O during MOSAiC

To identify key factors driving the interaction between atmosphere and snow, we compare $\delta^{18}$O in surface snow ($\delta^{18}O_{Snow}$) with a continuous $\delta^{18}$O times series in water vapour ($\delta^{18}O_{Vapour}$). Both datasets were averaged to monthly means and compared to aggregated weekly precipitation amounts (Figure 2). The number of snow samples collected at the surface during each
month varies from 7 samples in October 2019 and August 2020 to 39 samples in February 2020. The majority of samples (more than 65 %) were collected between October 2019 and May 2020. In summer, a reduced number of surface snow samples were taken due to unstable sea-ice conditions and extensive melting processes, with snow predominantly present in the form of slush and development of a surface scattering layer (Macfarlane et al., 2023b).

Both, $\delta^{18}O_{Snow}$ and of $\delta^{18}O_{Vapour}$ show a distinct seasonal cycle (Figure 2), although PS is passively moving southwards from the central Arctic between October 2019 and July 2020 and then travels to the central Arctic Ocean in August 2020 (Figure 3a). Median $\delta^{18}O_{Snow}$ values reach a maximum around -20 ‰ in October to December 2019, decreasing to the absolute minimum of ca. -33 ‰ in March 2020. Then, median $\delta^{18}O_{Snow}$ values rise to the absolute maximum of ca. -11 ‰ in July 2020, with a slight decrease to ca. -15 ‰ in August and September 2020, where samples mostly consisted of surface scattering layer.
The summer months show the largest variability in $\delta^{18}O_{Snow}$, whereas the winter months (November and December 2019) and some months in the transitional seasons (March and September 2020) vary less. Water isotopes in vapour display generally lower values, decreasing slightly from a maximum in October 2019 ($\delta^{18}O_{Vapour}$ of ca. -37 ‰) to the absolute minimum $\delta^{18}O_{Vapour}$ in March 2019 ($\delta^{18}O_{Vapour}$ of ca. -43 ‰). Afterwards, the median $\delta^{18}O_{Vapour}$ values increased continuously to the absolute maximum in August, 2020 ($\delta^{18}O_{Vapour}$ of ca. -18 ‰). In September 2020, when PS moved to a more northward position,
$\delta^{18}O_{Snow}$ and $\delta^{18}O_{Vapour}$ both decreased slightly.

Despite the general similar seasonal evolution of $\delta^{18}O_{Snow}$ and $\delta^{18}O_{Vapour}$, there is a noticeable offset in their trends, especially in winter. In this period, the $\delta^{18}O_{Vapour}$ values are rather stable and range from -35 to -45 ‰, while $\delta^{18}O_{Snow}$ shows a continuous decrease from about -20 to -35 ‰ from October to March 2019. Throughout the expedition, the median $\delta^{18}O_{Vapour}$ remains
lower than $\delta^{18}O_{Snow}$. This isotope difference between vapour and snow median $\delta^{18}$O values reaches 16 ‰ in October 2019, increasing to a largest offset of 24 ‰ in November (Table S1), then decreasing from 19 ‰ in December 2019, to 15 ‰ in January to March 2020.  The minimal isotope difference between vapour and snow ranges between 3 and 8 ‰ between May and August 2020, and increases to 11 ‰ in September 2020. This divergence between $\delta^{18}O_{Snow}$ and $\delta^{18}O_{Vapour}$ in winter and convergence in summer suggests a seasonal behaviour, where the isotopic composition of surface snow responds differently
with respect to vapour during specific periods, possibly influenced by distinct environmental factors.

The aggregated weekly precipitation amounts exhibit higher values in the summer months compared to winter. In months with consistently higher precipitation amounts, particularly from May onwards, a notable narrowing of the $\delta^{18}$O difference between



vapour and snow is observed. Hence, during periods of higher precipitation, the isotopic signals in snow and vapour become
more closely aligned.



**Figure 2 Monthly δ18O variations for vapour (pink) and surface snow (blue) during the MOSAiC expedition (October 2019 to September 2020). Monthly median values of δ18O are indicated by the central line within each boxplot, while the box edges correspond to the 25th and 75th percentiles. Whiskers extend to the non-outlier maximum and minimum values, with outliers depicted as plus symbols. The number of samples analysed for each month is given in the boxes. The weekly aggregated precipitation amounts, derived from the ERA5 reanalysis, are depicted as grey bars in a time series.**





### 3.3 Event-based precipitation isotopes

Figure 3a shows the route taken by PS in the Arctic during the MOSAiC expedition commencing at around 85 °N, 128 °E and heading towards the Fram Strait. This coloured track highlights the passive movement of the vessel by Transpolar Drift as it drifted with the Arctic ice pack, whereas transfer phases are given as grey lines. Figures 3b and 3c show the isotopic compositions of $\delta^{18}O$ and *d-excess* in event-based precipitation, sampled at two continental stations: Samoylov Island and Ny-Ålesund (PAPIN network). The former is located on the northern coast of Siberia, whereas the latter is located on Svalbard at the eastern entrance to the Fram Strait. The purpose of comparing snow samples collected during the MOSAiC expedition with precipitation event data from these stations is to better understand and disentangle spatial and temporal variations in the isotopic dynamics of Arctic snow, including how it is deposited and altered afterward. This knowledge is particularly valuable due to the absence of event-based precipitation sampling during the MOSAiC expedition. Long-term precipitation isotope data from Leroy-Dos Santos et al. (2020) for Ny-Ålesund and Bonne et al. (2020) for Samoylov Island are included to provide multi-year background information. The $\delta^{18}O$ values at Samoylov Island exhibit a clear seasonal cyclicity, as observed in both historical and MOSAiC-year records. During the winter months, there is a substantial decrease in $\delta^{18}O$ values from about -20 ‰ to -35 ‰. As temperatures increase in spring, the $\delta^{18}O$ values gradually rise to around -15 ‰ or higher. In contrast, Ny-Ålesund precipitation isotope data displays little fluctuation in $\delta^{18}O$ over the seasons, with $\delta^{18}O$ values stable around -10‰.

Both stations provide a wide range of *d-excess* values throughout the year. Samoylov Island's median *d-excess* ranges around +10 ‰, with individual measurements down to -40 ‰. On the other hand, Ny-Ålesund exhibits larger variations, with median *d-excess* values fluctuating between around +5 ‰ in October and +25 ‰ in December, and occasionally reaching maxima as high as +35 ‰. During the period from May to September, both stations consistently show similar *d-excess* values that fall within the range of +10 to -10 ‰.



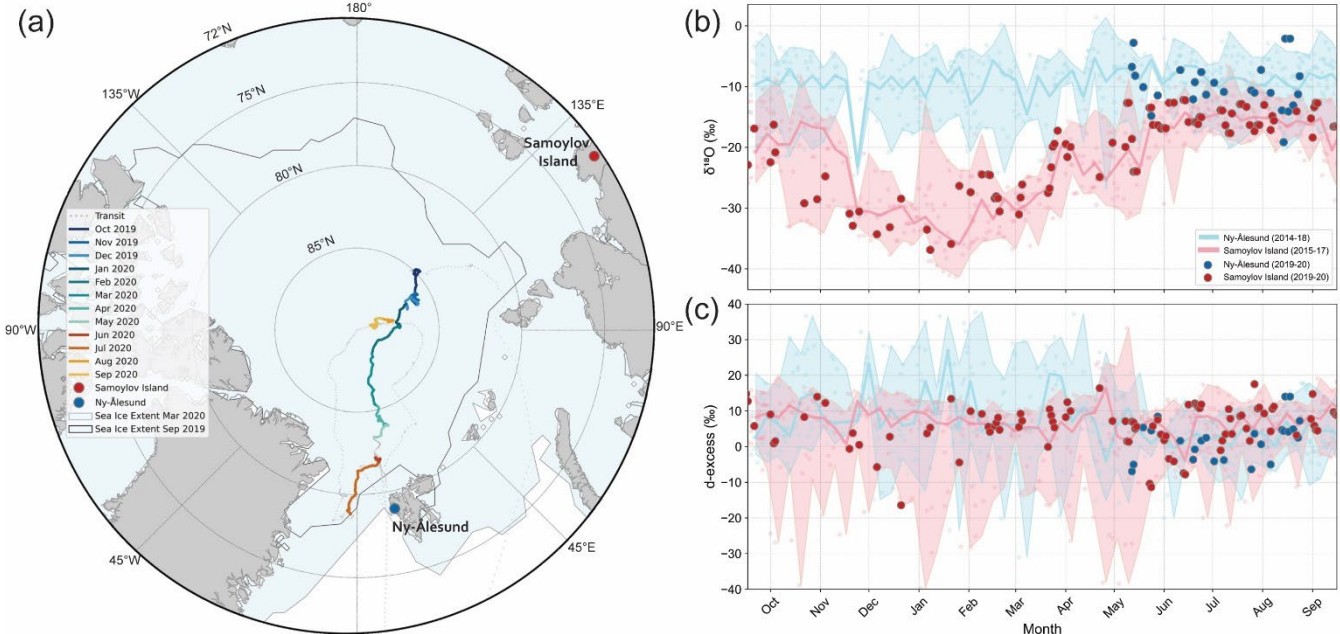

**Figure 3 Polar projection of the Arctic Ocean with PS's track during the MOSAiC expedition color-coded by months; transit periods**
**are represented by grey dots. Locations of event-based precipitation sampling on Samoylov Island and at Ny-Ålesund stations are indicated. Sea-ice extent at annual maximum and minimum (MOSAiC year) is depicted in light blue shading. b) and c) present $\delta^{18}O$ and *d-excess* measurements of sampled precipitation events for an annual cycle from October to September for both stations. Light blue and light red shadings illustrate the range of isotope values (min to max) from Ny-Ålesund and Samoylov Island, respectively, while the corresponding weekly median values are depicted as thicker blue and red lines. Dots overlaying the line graphs denote**
**precipitation event samples collected during the MOSAiC year from Ny-Ålesund (blue) and Samoylov Island (red).**

There are similarities and dissimilarities between monthly mean $\delta^{18}O_{snow}$ obtained during the MOSAiC expedition and precipitation samples recorded at both stations. Generally, the MOSAiC surface snow $\delta^{18}O$ data shows a larger monthly variability than the precipitation data at both stations. The $\delta^{18}O_{snow}$ data shows a clear seasonal cyclicity similar to that of Samoylov Island precipitation, not visible in the Ny-Ålesund multi-annual dataset (Figure 4a). In some months, a distinct offset

between the MOSAiC surface snow and Samoylov Island is observed. For instance, in November and December 2019, the median $\delta^{18}O$ in precipitation at Samoylov Island is depleted by about 12 ‰ relative to that of surface snow. This offset decreases in January to -6 ‰, whereas from February to April the offset is positive from 2 to 4 ‰. From May to September 2020, generally wider $\delta^{18}O$ ranges of were recorded both at the stations and for surface snow. Generally, the median $\delta^{18}O_{Snow}$ exhibits a negative offset compared to Ny-Ålesund data, and, except for August, consistently negative offsets with Samoylov

Island data, ranging from -2.6 ‰ (May) to -15 ‰ (July). The $\delta^{18}O$ interquartile ranges for surface snow and Samoylov Island overlap between January and September, whereas for Ny-Ålesund between June and August 2020, only.

The *d-excess* from MOSAiC surface snow shows significant month-to-month variability, ranging from slightly negative values in summer to over +20 ‰ during the winter months. The median values from October 2019 to May 2020 are typically higher



than +10‰. Consequently, a clear seasonal *d-excess* cyclicity in surface snow is visible, characterised by opposite trends between *d-excess* and $\delta^{18}O_{Snow}$ values (Figure 4). A noticeable offset between the median *d-excess* in surface snow and that of Samoylov Island is observed during winter, with Samoylov Island's *d-excess* being up to 15 ‰ lower (December, January) than that of surface snow. This offset gradually decreases towards zero by June (-2 ‰) and reverses up to 9 ‰ (July, August) in the summer months. Hence, seasonal *d-excess* cyclicity in Samoylov Island precipitation is not visible in the dataset, especially in winter. However, *d-excess* values of surface snow overlap within their interquartile ranges with that of Samoylov Island precipitation, from June to November.

The MOSAiC surface snow *d-excess* values often closely match those recorded at Ny-Ålesund. Ny-Ålesund *d-excess* values have a narrow distribution in summer, mostly ranging between +5 ‰ and +15 ‰. However, the median *d-excess* of Ny-Ålesund is up to +17 ‰ lower than the surface snow median (in May 2020). Between June and September, Ny-Ålesund *d-excess* values overlap within their interquartile ranges with that of surface snow. The variance in Ny-Ålesund's *d-excess* values also increases during winter (Figure 4b).





**Figure 4 Monthly mean of isotope data from Arctic surface snow and continental precipitation (for the MOSAiC Year). Box plots provide a statistical summary of the monthly δ¹⁸O (a) and *d-excess* (b) variability of surface snow compared to precipitation events at Samoylov Island and Ny-Ålesund stations, from the Pan-Arctic Precipitation Isotopes Network (PAPIN). In each boxplot, the medians are indicated by vertical lines, the box boundaries represent the interquartile ranges, and the whiskers extend to the minimum and maximum values for each month.**



## 4 Discussion

In this study, we explore the seasonal and spatial dynamics and the interplay between deposition and post-depositional processes of snow on the Arctic sea-ice, using a full year of discrete snow isotope measurements from MOSAiC. The isotopic composition of precipitation, particularly snow, is a valuable indicator for studying atmospheric processes and interpreting climatic conditions in polar regions (Reinwarth et al., 1985; Petit et al., 1991; Steen-Larsen et al., 2014; Ala-Aho, 2021; Wahl et al., 2022; Zuhr et al., 2023). We compare the snow data to continuous vapour isotope measurements to gain insights into the predominant snow-atmosphere exchange processes.

### 4.1 Snow profiles: meteoric water origin and sea-ice-snow interactions

The changes in the isotopic composition of snow profiles reflect the contribution of relatively depleted meteoric water, and snow-sea-ice dynamics characterised by a relative enrichment in $\delta^{18}O$ (Figure 1a). Starting in January 2020, snow height increases, eventually reaching a height of 80 cm between March and May 2020 in ridge positions. During snow accumulation, a differentiation occurs within the snow cover: the upper layer becomes more depleted in $\delta^{18}O$ and reveals higher *d-excess* values, which might indicate recent snowfall while the lower layer becomes more enriched in $\delta^{18}O$ with lower than median *d-excess* values, which is presumably caused by snow interacting with the sea-ice surface through diffusion of sea-ice (Macfarlane et al., 2023a; Mellat et al., 2024). This lower layer in the snowpack is particularly noticeable when snow covers are thin (Figure 1a). The contrast between fresh meteoric snow and snow influenced by sea-ice diffusion throughout the months of January to May leads to pronounced isotopic gradients (Figure 1). These results may be important for refining modelling strategies used in regional and global atmosphere models with isotope diagnostics (Brunello et al., 2023). They show that in certain phases, e.g. the early winter when snowfall is minimal, isotopic exchanges between the atmosphere and Arctic sea-ice covered surfaces tends to occur directly between the vapour and the sea-ice rather than with snow on the sea-ice.

Between June and September 2020, as temperatures approach 0 °C, the initial thaw causes the formation of melt ponds onto the sea-ice. These ponds deepen progressively until they penetrate through the sea-ice, facilitating mixing with seawater (Nomura et al., 2023). The early stages of melting are characterised by a shallow and wet layer of snow that covers the sea-ice, influenced by two distinct isotopic signatures. At first, meltwater percolation starts from the surface, resulting in the preferential melting of fresh snow. During this process, meltwater percolates through the snowpack, diminishing the variation in its isotopic composition (Moran and Marshall, 2009). Such processes result in increasingly homogeneous isotope profiles within the snowpack, as meltwater reduces snow height and isotopic variability (Figure 1). Disequilibrium fractionation due to melting may contribute to further enrichment of $\delta^{18}O$ in the snowpack with a tendency towards lower $\delta^{18}O$ for the initial meltwater, which then shifts to higher $\delta^{18}O$ values as partial melting continues (Friedman et al., 1991; Clark and Fritz, 2013). This leads to an isotopic enrichment of $\delta^{18}O_{Snow}$, generally associated with higher *d-excess* values, when initial meltwaters




drain into melt ponds, leads, and the ocean (Nomura et al., 2023). By August and September 2020 melting processes cause a relatively δ$^{18}$O-enriched, homogenised snowpack. This phase reflects the interplay between melting dynamics and isotopic composition, responsible for alterations of isotopes at the Arctic snow and sea-ice interface in this period of the year.

## 4.2 Linkage between MOSAiC surface snow and atmospheric water vapour

The δ$^{18}$O (Figure 2) reveals distinct patterns of match and mismatch between the isotopic signatures of snow and vapour during the MOSAiC campaign. We see an initial offset between δ$^{18}$O$_{Snow}$ and δ$^{18}$O$_{Vapour}$, which decreases during winter and aligns from May to September 2020 (Figure 2). This transition throughout the MOSAiC-year highlights (1) the relationship between the intensity and intermittency of precipitation and (2) the transfer of isotopic signatures between the atmospheric water vapour and surface snow.

October to December 2019 is the period when the largest mismatch between snow and vapour δ$^{18}$O values occurs (Figure2). This offset can be attributed to two primary factors: firstly, whereas the vapour time series reflects real-time conditions, the snow cover relates to one or several previous precipitation events and a few days, weeks, or even months might have occurred between initial precipitation and sampling. Hence, the snow might carry an inherited isotopic signature from older winter snow as seen in other snow studies from Greenland (Zuhr et al., 2021), which is not necessarily related to the vapour at a given time. Secondly, the influx of warm air during autumn (Figure 7) often brings additional moisture and heat to the region, generally characterised by isotopically-enriched Atlantic moisture (Brunello et al., 2023), which would lead to higher δ$^{18}$O values in snow. Similar values and a relatively close range of *d-excess* in snow and vapour during this period (Figure S1) could be either interpreted as an indicator for similar moisture sources for snow and vapour, or indicate little influence of secondary processes taking place after deposition of snow.

From January and May 2020, a phase of discrepancy between δ$^{18}$O$_{Snow}$ and δ$^{18}$O$_{Vapour}$ (Figure 2) as well as between *d-excess* in vapour and snow is observed (Figure S1). During this period, there were substantial differences between the δ$^{18}$O$_{Snow}$ and δ$^{18}$O$_{Vapour}$, of up to 24 ‰ (Figure 2). This period corresponds to the most extreme cold and dry period of winter, when PS was initially located in the Siberian sector of the Arctic Ocean, then drifting into the Atlantic sector (Figure 3a) in March 2020. These offsets may imply a deviation from equilibrium fractionation, suggesting kinetic isotopic fractionation between vapour and snow.

During the period from May to August 2020, we observed minimal isotopic differences between snow and vapour δ$^{18}$O (Figure 2) as well as between *d-excess* of snow and vapour. This phase is likely characterised by conditions favourable to equilibrium fractionation between vapour and snow, such as higher temperatures and humidity, and increased precipitation rates. These conditions enable a more effective transfer of isotopic signals, allowing for a direct and efficient exchange between vapour and snow (Beria et al., 2018). This is particularly evident when the PS is positioned within the Atlantic sector of the Arctic



Ocean. During these periods, there seems to be a direct contact and clear interaction between atmospheric water vapour and snow surfaces, such as the transfer of moisture to the snow surface through vapour deposition (Casado et al., 2016).

The observed offsets between $\delta^{18}O_{Snow}$ and $\delta^{18}O_{Vapour}$, as well as related *d-excess* values, allow for the categorization of the development of the Arctic snow cover into three distinct stages: (1) an early winter phase, with a clear offset in $\delta^{18}O$, but not

in *d-excess*, (2) the late winter phase characterised by extremely cold temperatures and low humidity, where the disequilibrium between vapour and snow is most pronounced, and (3) the spring to summer phase, where the difference is significantly smaller, suggesting conditions closer to equilibrium fractionation, as well as impact of melting, mixing with relatively enriched sea-ice, and refreezing. Furthermore, the offset experiences another rise in September 2020 when the PS travels back to the central Arctic (87 °N), where PS passed during February 2020, half a year before. This marks the end of

the annual cycle, essentially reverting to the autumn situation and closing the seasonal cycle observed during the MOSAiC-year.

### 4.3 Impacts of precipitation amounts and air temperature on snow-vapour isotopic correlations

The presence of a moderate to weak correlation between the $\delta^{18}O$ values of snow and vapour reflects the theoretical concept of equilibrium fractionation ($R^2 = 0.23$, Figure 5b). If considering major precipitation events only, here defined as those

exceeding 2 mm per day, the correlation between $\delta^{18}O_{Snow}$ values strengthens ($R^2 = 0.37$, Figure 5b). We assume that the observed stronger correlation coefficients are likely due to higher humidity levels during these precipitation events, and a faster and more direct transfer of isotopic signatures between vapour and snow. In this scenario, more equilibrium fractionation is expected and the atmospheric residence times are shorter. Hence, precipitation amount has an impact on $\delta^{18}O$ signatures, especially during the summer months in the Arctic. The increase in precipitation during this time likely influences the

convergence pattern between vapour and surface snow $\delta^{18}O$ in the warm season. However, the correlations between $\delta^{18}O_{Snow}$ and $\delta^{18}O_{Vapour}$ remain low, suggesting the persistent, overlaying impact of post-depositional processes decoupling the two datasets (Figure 5). During summer months, characterised by the absence of fresh snowfall, sublimation and melting significantly enrich the snow's isotopic composition. Such a mechanism explains the relatively higher $\delta^{18}O$ values recorded from October to December 2019, hinting at an inheritance of the prior summer's isotopic signature persisting into the colder

months.

The relationship between $\delta^{18}O$ values in snow and vapour with air temperature further elucidates the seasonal dynamics. $\delta^{18}O_{Vapour}$ exhibits a strong correlation with temperature ($R^2 = 0.83$, Figure 6a), in agreement with the theoretical framework (Dansgaard, 1964). In contrast, the correlation of $\delta^{18}O_{Snow}$ with temperature is weaker ($R^2 = 0.26$), highlighting the significant





impact of post-depositional processes. Thus, Arctic snow does not merely reflect the conditions at the time of deposition but
       is also a product of ongoing physical alterations within the snowpack.

       Throughout the different stages of the expedition, the relationship between average daily snow and vapour $\delta^{18}O$ correlations
       with average daily air temperatures revealed distinct seasonal dynamics (Figure 6). During the initial phase of the expedition,
from October to December 2019, $\delta^{18}O_{Snow}$ values appeared to be unaffected by temperature fluctuations, whereas $\delta^{18}O_{Vapour}$
       showed a predictable reaction to temperature changes with a slope of 0.69 ‰/°C, aligning with expectations for vapour's
       sensitivity to temperature. During the winter months from January to May 2020 there is a slight increase in snow's temperature
       response, indicated by a slope of 0.18 ‰/°C. However, a broad range of $\delta^{18}O_{Snow}$ values persisted within similar temperature
       conditions, suggesting a complex interplay of factors influencing snow's isotopic composition. The summer period exhibited
the most pronounced response in $\delta^{18}O_{Vapour}$ to abrupt temperature rises, with a high correlation ($R^2 = 0.72$, Figure 6d) and a
       steep T-$\delta^{18}O$ slope of 1.98, likely reflecting the increased relative humidity during these warmer months. Conversely, $\delta^{18}O_{Snow}$
       values during the summer showed an anticorrelation with air temperature, varying between -7 and +1 °C. This period was
       characterised by a bifurcation in $\delta^{18}O_{Snow}$, with nearly half of the samples being significantly depleted (-30 to -15‰) and the
       other half being enriched (-15 to 0 ‰). The depleted samples, predominantly collected from ridges in June and July, presented
a mixed signal of snow redistributed across the sea-ice cover, accumulating at ridges where conditions favoured snow
       accumulation up to 80 cm. These ridge samples, particularly within the upper 40 cm, largely retained the re-distributed snow
       isotopic signature, indicating minimal alteration from deposition to collection. Nonetheless, the correlation between T and
       $\delta^{18}O$ of deposited snow on the sea ice collected during MOSAiC has been found to be weak (Mellat et al., 2024).

       Conversely, the higher $\delta^{18}O_{Snow}$ values found in samples from flat ice during this period suggest the impact of melting and
refreezing cycles, especially within the surface scattering layer linked to surface snow metamorphism. This process has been
       addressed by studies conducted by Steen-Larsen et al. (2014) and Casado et al. (2018). The dynamic nature of snow-
       atmosphere interactions can be noticed by the rapid changes in the isotopic composition of the snow surface which are reported
       to happen on a sub-daily timescale, driven by sublimation and vapour deposition (Ritter et al., 2016; Wahl et al., 2021). This
       effect is especially relevant during periods without precipitation, where sublimation can enrich the remaining snow in heavier
isotopes (Wahl et al., 2022). This metamorphism is not confined to the MOSAiC surface snow but is also evident across the





snowpack sampled during summer, which displayed more enriched profiles compared to the winter, where a gradient from low to high $\delta^{18}O$ signals in snow profiles was observed (Figure 1).

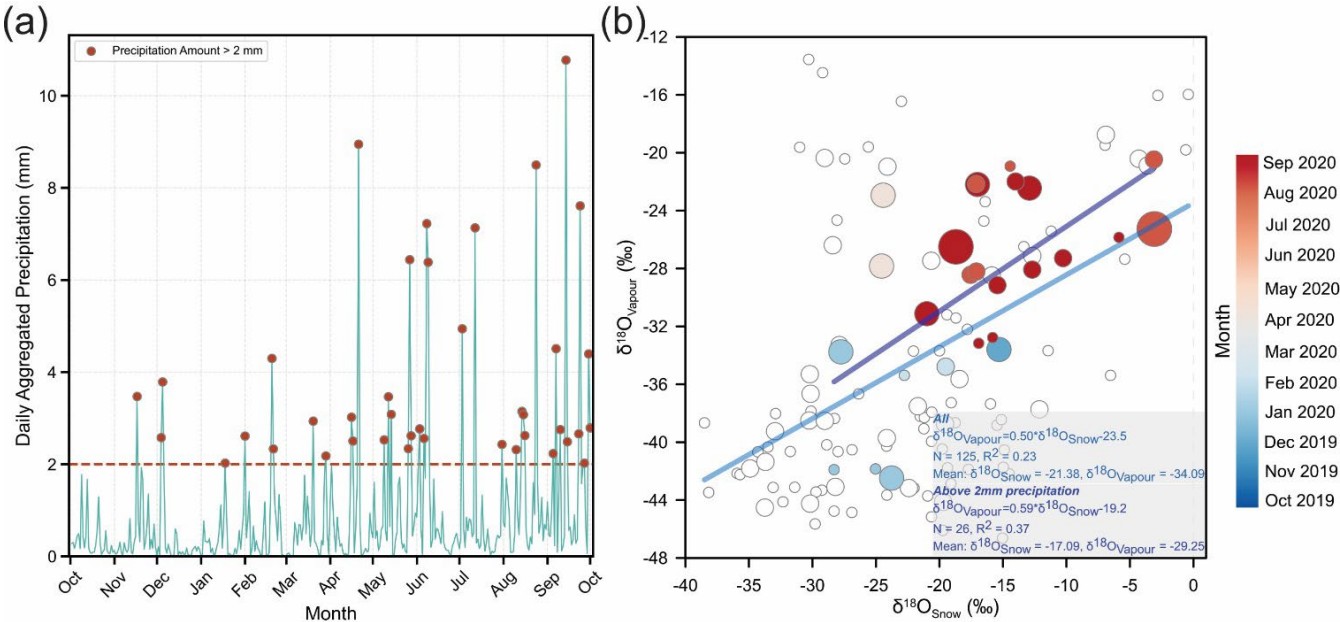

**Figure 5 Comparative Analysis of Precipitation and Isotopic Composition. a) The left panel displays the daily aggregated precipitation over a year (reanalysis data from ERA5), with the horizontal dashed line indicating a threshold of 2 mm for significant precipitation events, highlighted by red circles. b) The right panel presents a scatter plot correlating the $\delta^{18}O$ values of snow ($\delta^{18}O_{Snow}$) with those of vapour ($\delta^{18}O_{Vapour}$). Each point represents a measurement, with the colour gradient reflecting the progression of months from October 2019 to September 2020. Two distinct regression lines are shown: the light blue line represents all surface snow samples (N = 125), while the dark blue line corresponds to surface snow samples collected within 5 days after precipitation events of more than 2 mm (N = 26). The corresponding regression equations, coefficient of determination ($R^2$), and mean $\delta^{18}O$ values for snow and vapour are provided for both datasets. The size of each point is proportional to the amount of precipitation on the corresponding day.**







**Figure 6 Temperature Dependence of Isotopic Ratios in Snow and Vapour.** The figure illustrates the relationship between air temperature (T (°C)) and the mean daily $\delta^{18}O$ surface snow ($\delta^{18}O_{Snow}$) and vapour ($\delta^{18}O_{Vapour}$) during (a) MOSAiC expedition from October 2019 to September 2020. Data points are differentiated by the period of the year they represent: circles for October to December 2019, triangles for January to May 2020, and squares for June to September 2020. (b) represents data from October to December 2019, (c) from January to May 2020, and (d) from June to September 2020. Each panel illustrates the corresponding linear regression analyses for $\delta^{18}O_{Snow}$ (blue lines) and $\delta^{18}O_{Vapour}$ (red lines) within each period. The shaded areas around each regression line indicate the 95% confidence intervals.

## 4.4 Comparison of MOSAiC surface snow with continental precipitation

The PS's transit between Samoylov Island in Northern Siberia and Ny-Ålesund Station on Svalbard offers the opportunity to examine their isotopic signatures with respect to the starting and endpoint of the MOSAiC expedition. Our observations indicate a distinction in the seasonal $\delta^{18}O$ patterns between Samoylov Island and Ny-Ålesund. Samoylov Island



shows a clear annual cyclicity, with variations in δ¹⁸O values that reflect the wide temperature range (-45 °C to +25 °C) typical of the continental environment, with pronounced winter minima and summer maxima in δ¹⁸O values. On the other hand, Ny-Ålesund, being close to the Atlantic Ocean as the main source of moisture, exhibits moderate seasonal δ¹⁸O variations and a lower annual temperature range (-12 °C to +6 °C). The lack of a clear seasonality at Ny-Ålesund is related to the influence of

specific climatic conditions in the region with a year-round influx of isotopically enriched Atlantic moisture at this station (Boisvert et al., 2023). Colder ambient conditions in Siberia are reflected in the consistently lower δ¹⁸O values at Samoylov Island compared to Ny-Ålesund throughout the year. For the *d-excess* values, Ny-Ålesund data display slightly higher variability in comparison to Samoylov Island. The decrease in variability observed at Samoylov Island over the summer for both δ¹⁸O and *d-excess* is not visible at Ny-Ålesund. This observation suggests that the *d-excess*, although widely used as a

proxy for identifying moisture sources for precipitation, may not effectively differentiate between moisture originating from proximal versus distal sources with similar atmospheric humidity and temperature.

We also conducted a comparative analysis of these precipitation data with their respective long-term climatology (Leroy-Dos Santos et al., 2020; Bonne et al., 2020). The precipitation data collected at Samoylov Island during MOSAiC closely aligns

with the long-term median, suggesting that the year was typical in terms of δ¹⁸O and *d-excess* values, although, the occurrence of lower δ¹⁸O values in November 2019 (Figure 3b) may indicate earlier arrival of colder conditions. Also, the δ¹⁸O values in precipitation measured Ny-Ålesund from May to September 2020 align with the median values of long-term δ¹⁸O recorded for the same period. The consistency in Ny-Ålesund's *d-excess* values between long-term observations and the MOSAiC-year suggests that the environmental conditions, specifically humidity and temperature ratios, adhered to historical summer trends.


When comparing the MOSAiC snow dataset to the simultaneous precipitation data at the coastal stations, we observe an early decrease in winter snow δ¹⁸O values compared to that in the Samoylov Island data. This two-month delay in the isotopic signal's response to atmospheric conditions at Samoylov Island, observed by a later minimum in March 2020, hints at an indirect relationship between precipitation and its integration into the snowpack. The observed temporal difference indicates

that it takes roughly two months for the snowpack to align with the δ¹⁸O values of precipitation observed at Samoylov Island. Therefore, this suggests that alterations in continental precipitation are unlikely to immediately correspond to changes in the snow on Arctic sea ice. This observation could also suggest different winter precipitation patterns or moisture sources between Samoylov Island and central Arctic Ocean. However, during October and November 2019, Samoylov Island precipitation and MOSAiC snow have similar mean *d-excess* (ca. 10 ‰, Figure 4b) but these values start to diverge as PS moves further away

from Samoylov Island. This means that moisture sources are slowly changing between the two observatories. This offset varies in the later months until June and July, when the Arctic region is in mostly under the influence of Atlantic moisture transported into the Arctic, leading to lower *d-excess* values both in Samoylov Island and Ny-Ålesund precipitation (as reported earlier by Mellat et al. (2021)), as well as MOSAiC snow.





**4.5 Depositional and post-depositional processes imprints on surface snow isotopes**

The response of $\delta^{18}O$ in both surface snow and vapour to meteorological conditions points out to interactions between air temperature and humidity as main depositional factors. The variations in the $\delta^{18}O_{Vapour}$ during warm air intrusions, like those that occurred in November 2019 and April 2020 (Rinke et al., 2021), show that the vapour responds to changes in air temperature and relative humidity (Brunello et al., 2023). However, $\delta^{18}O_{Snow}$ appears to be less sensitive to immediate atmospheric conditions. Although warm air brought into the Arctic is expected to contribute precipitation to the snow profiles,

the isotopic signature of $\delta^{18}O_{Snow}$ does not show a clear impact from such additions (Figure 7), suggesting that post-depositional processes as well as wind redistribution might impact its isotopic signature.

Between December 2019 and March 2020, a gradual depletion in $\delta^{18}O_{Snow}$ was observed (Figure 7a), primarily driven by a concurrent slight decrease in temperature levels (Figure 7b). However, relatively constant cold conditions during this period

also indicate that sublimation may have contributed to this depletion trend. This is supported by the mostly (72 % of the total period) negative LHF observed during this time (Figure 7d). Sublimation introduces another level of complexity to interpreting the deposited snow isotopic signature. It has been found that the longer residence time of snow on the surface, which facilitates isotopic homogeneity of individual ice grains through solid-phase diffusion, may enrich the isotopic composition, including $\delta^{18}O$ and *d-excess* (Zuhr et al., 2023). Thus, in contrast with the previous assumptions of layer-by-layer sublimation without

isotopic fractionation (Ambach et al., 1968; Dansgaard, 1973), sublimation may play a key control on the isotopic composition of snow in polar regions. The porosity of snow allows sublimation, leading to an influence on both the $\delta^{18}O$ and *d-excess* (Casado et al., 2021).

High wind velocities and sea-ice's aerodynamic properties cause significant snow redistribution, with large amounts of snow

mass lost through sublimation when snow is in the air (Wagner et al., 2022). Moreover, the flattening of the snow surface and reduction in surface roughness due to wind-driven erosion can influence sublimation rates and the efficiency of vapour exchange between the snowpack and the atmosphere (Zuhr et al., 2021). This indicates the dynamic nature of snow distribution and its effects on isotopic composition, highlighting the role of physical and thermodynamic changes in altering isotopic records. During the winter period, there was a significant redistribution of snow, which occurred 67 % of the time (WS > 3

m/s, Figure 7e). The mixing and redistribution of snow on sea-ice likely homogenised the $\delta^{18}O_{Snow}$ signal of the snow deposited over the sea ice. This process led to a shift toward a depleted isotopic composition in the top layer of the snow, as snow with lower $\delta^{18}O$ values from meteoric sources was spread across the surface, likely also responsible for the low T- $\delta^{18}O$ correlation coefficients in surface snow (Figure 6). This mixing effect is also apparent in the snow profiles collected from the ridges, where the upper layers exhibit a homogeneous $\delta^{18}O$ signal, in contrast to the flat ice samples (Figure 1a).




June and July 2020, experienced the highest air temperatures throughout the expedition and near-saturation levels of humidity. Although sublimation occurred less frequently, representing just 12% of times (LHF < 0 W/m$^2$), snow redistribution was notably higher, accounting for 44% of the period. Despite this, the snow on the sea ice was predominantly identified as wet, comprising a mix of slush, making its redistribution less probable. Therefore, snow redistribution plays a more significant role in altering the isotopic signature of surface snow on the sea ice during winter than in summer. Over this warmer time, it is noteworthy that half of the collected samples displayed a $\delta^{18}O_{Snow}$ influenced by the real-time meteorological conditions, resulting in more enriched values ranging from 0 to -15 ‰. The remaining half of the samples exhibited a signal more typical of the winter surface snow, with values ranging from -20 to -30 ‰ (Figure 7a).



**Figure 7 Seasonal fluctuations in meteorological and isotopic data captured during the MOSAiC expedition. a) depicts daily average δ¹⁸O isotopic ratios in surface snow (blue dots) and the daily average vapour (red line). b) illustrates air temperature at 2 m above ground level (T, °C) on an hourly basis, and c) the hourly relative humidity (RH, %). d) shows the latent heat flux (LHF, W/m²) measured hourly. Positive LHF values suggest evaporative loss and negative values deposition, revealing intermittent events across the seasons. The bottom graph shows the hourly wind speed (WS, m/s), applying a 3 m/s threshold to denote significant snow movement as suggested by Wagner et al. (2022); wind speeds surpassing this threshold are highlighted in light blue. The period of warm air intrusion event identified during the expedition (Rinke et al., 2021) is shown as grey bars on all panels. The meteorological data are extracted from the MET tower observations during MOSAiC expedition (Cox et al., 2023).**



## 5 Conclusions

This study addresses the link between the deposition and post-depositional processes that impact the stable water isotope signal in snow on Arctic sea-ice, using a unique dataset from the MOSAiC expedition. The analysis suggests that the isotopic composition of Arctic snow is only partially controlled by depositional processes, and instead is heavily influenced by post-depositional processes. The use of continuous vapour isotope measurements alongside discrete snow samples represents a methodological advancement, allowing for a better understanding of the snow-atmosphere interaction and the associated processes affecting isotopic signatures in the Arctic snow.

Through the analysis of discrete snow isotope measurements during an entire year, we found that snow on Arctic sea-ice displays a distinct stratification, with depleted $\delta^{18}$O and higher *d-excess* in the upper layer indicating meteoric snowfall contribution, and enriched $\delta^{18}$O and lower *d-excess* values in the bottom layer due to interactions with underlying sea-ice. This insight is crucial for accurately modelling isotopic exchanges, especially in conditions of minimal snowfall, to differentiate between meteoric and sea-ice vapour sources in the Arctic.

Our study identified three main phases in the isotopic composition of Arctic snow cover throughout the year. Each phase is characterised by distinctive connections between the $\delta^{18}$O values and *d-excess* of snow and vapour, demonstrating the dynamic nature of Arctic snow-sea-ice interactions:

(1) Autumn and early winter is characterised by a $\delta^{18}$O offset with relatively enriched snow, and consistently positive *d-excess* in snow, reflecting a lag in the snow isotopic adaptation to precipitation and the impact of isotopically-enriched Atlantic air. During this phase, $\delta^{18}$O$_{Snow}$ remains relatively unaffected by air temperature shifts, whereas $\delta^{18}$O$_{Vapour}$ shows expected temperature correlations, highlighting the significant role of post-depositional processes on the sea ice in altering $\delta^{18}$O$_{Snow}$.

(2) Cold months of winter are marked by a pronounced decoupling between vapour and snow under low humidity. During the winter, there was a slight rise in temperature response in snow, and a wide range of $\delta^{18}$O$_{Snow}$ values were recorded. This suggests that factors such as wind redistribution of deposited snow were involved.

(3) Transitioning into spring and summer, the gap between snow and vapour isotopic signatures narrows, suggesting a move towards equilibrium fractionation. This is accompanied by a noticeable link between warming temperatures, increased humidity, and a pronounced $\delta^{18}$O$_{Vapour}$ response. Surface snow differentiates into two categories: ridge snow, mirroring winter's depleted $\delta^{18}$O values, and flat ice snow, exhibiting enriched $\delta^{18}$O due to melt-freeze cycles and sea ice meltwater mixing. Additionally, this period witnesses intensified sublimation, leading to a further increase in $\delta^{18}$O content in surface snow.

Comparing the data from MOSAiC with the precipitation isotopes from Samoylov Island and Ny-Ålesund reveals that there are distinct seasonal isotopic patterns in these locations. Following the initial alignment, the $\delta^{18}$O values of



the MOSAiC snow deviate from those observed at Samoylov Island for two months (November and December). This suggests that there are different meteorological and post-depositional conditions between Samoylov Island and the central Arctic Ocean. During the summer, all three sites exhibit similar enriched $\delta^{18}O$ ratios and lower *d-excess* values, suggesting that they are all influenced by the same source of warm moisture that affects the Arctic region.


Future research should aim at improving the temporal resolution of snow isotope measurements to more precisely capture isotopic variability. There is a pressing need for detailed, short-interval measurements of snow isotopes and their physical properties, ideally on an hourly to daily basis. This would allow to better address sublimation, a phase transition known to significantly modify Arctic snow surface. Additionally, the deployment of continuous vapour isotope measurements in closer

proximity to the snow surface could provide a more detailed understanding of the local versus advective moisture contributions to the snow isotopic composition. These methodological improvements could significantly refine our understanding of the snow-atmosphere interactions and their representation in climate models.



**Data accessibility statement:**

All data in this manuscript are publicly available from online repositories. The data can be found under the following
references:

- Stable water isotopes of snow during MOSAiC expedition (Mellat et al., 2022).
- Snow pit raw data collected during the MOSAiC expedition (Macfarlane et al., 2022).
- Continuous near-surface atmospheric water vapour isotopic composition from Polarstern cruise PS122-1:5 (Brunello et al., 2022a, b, c, d, e)


**Author contributions:**

MM, ARM, HM, CFB, MW, and MS contributed to the conception and design. The acquisition of data was carried out by MM, ARM, HM, CFB, MW, MS, RD, SA, K-RM, and JMW. Analysis and interpretation of the data were performed by MM, ARM, CFB, HM, MW, MS, JMW, and RD. Drafting and revising the article was done by MM, ARM, CFB, MW, MS, RD,
SA, K-RM, JMW, and HM.

**Acknowledgements:**

We would like to express our gratitude to the International Multidisciplinary Drifting Observatory for the Study of the Arctic Climate (MOSAiC) with the tag MOSAiC20192020 for providing the opportunity to conduct this research and for producing
the data used in this report. Specifically, we would like to thank all people who were involved in this expedition of the Research Vessel Polarstern (Knust, 2017) and helped to collect the samples during MOSAiC in the 2019–2020 project (Project ID: AWI_PS122_00) (Nixdorf et al., 2021). We extend our sincere appreciation to all the persons involved in the expedition of the R/V Polarstern during MOSAiC in 2019–2020 (AWI_PS122_00). We also thank Mikaela Weiner and Andreas Marent for analysing the isotopic compositions ($\delta^{18}$O, $\delta^{2}$H) of the samples in the ISOLAB Facility at AWI in Potsdam.


**Funding statement:**

MM, CFB, MW, and HM are grateful for the support provided by the German Federal Ministry of Education and Research (project MOSAiC-CiASOM, grant number 03FO869A).

**Competing interests:**

All authors declare that they have no competing interests.



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
