# Peer review of "Arctic Surface Snow Interactions with the Atmosphere: Spatio-Temporal Isotopic Variability During the MOSAiC Expedition"

_EGUsphere, 2024_

## Author Comment (AC1)

**RC1: 'Comment on egusphere-2024-719', Lijun Tian, 21 Apr 2024**

General Comment:

The preprint, "Arctic Surface Snow Interactions with the Atmosphere: Spatio-Temporal Isotopic Variability During the MOSAiC Expedition," offers an in-depth analysis of the isotopic composition within Arctic snow and its dynamic relationship with the atmosphere. The research examined a robust dataset comprising 911 discrete snow isotope measurements, spanning the duration of an entire year throughout the MOSAiC expedition, complemented by continuous vapor isotope data. This research utilizes this unique dataset to explore the complex deposition processes and post-depositional changes affecting snow on Arctic sea ice. The study's findings are valuable for refining climate models and understanding the broader implications of Arctic climate change.

However, the discussion of the processes involved is too simple and elides much of the complexity involved, which could lead to overconfidence in the interpretation of isotopic signals. There needs to be a much clearer discussion of these processes and the potential errors they might introduce. Therefore, I recommend major revisions before it can be considered for publication.

Specific Comments:

1. There is significant bias ($\delta18O$-6.40‰; $\delta2H$-36.4‰) between the samples from dataset 1 and dataset 2. You attributed this systematic offset to the evaporative fractionation effects during storage of dataset 2 samples. I have never encountered such a substantial isotope difference between labs, particularly when the samples were stored in a low-temperature, sealed environment. In my opinion, there might be an issue with the standards and/or the correction process in the WSL lab. You could verify this by sending samples with known isotopic values to the WSL lab for re-analysis.

Thank you for your comment. We have done this. We measured 50 samples from dataset 2 in AWI Potsdam lab and had more or less similar values (with higher precision of course) as for measurements in the WSL lab. L160 (Methods): "*To ensure the robustness of the data, a comparison was conducted between the two datasets. This involved measuring a subset of 50 samples in both laboratories. By comparing the measurements of samples in both laboratories, we noticed a systematic offset between dataset 1 and 2 independent of the laboratory measurements.*" Thereby we concluded that the evaporative fractionation effects have caused the systematic offset between the two datasets as shown below in the comparison between the CiASOM and (uncorrected) SLF snow profiles from the same locations at the same time:

[Figure]

We discussed this correction for more than two months between the two groups in AWI Potsdam and SLF and tried different approaches to correct the dataset 2 and came down to this solution to be able to use both datasets.

2. There is no information on "snow ice" in this manuscript. When the snowpack on sea ice is heavy and thick enough to depress the top surface below sea level, a slush layer or slurry is formed through the mixture of seawater or brine and snow at the base of the snow cover. The surface flooding and snow-ice formation might exert influence on the snow isotopic signals. You should provide freeboard information and give a brief discussion on that. Two refs on snow-ice formation:

https://doi.org/10.1002/2017JC012865
https://doi.org/10.1017/aog.2020.55

This is not entirely the case in the Arctic Ocean, particularly in the context of the MOSAiC expedition. While snow-ice formation is more commonly observed in Antarctica due to significant snow accumulation on sea ice (often exceeding 70 cm), leading to the formation of snow-ice from the heavy weight of the snow, it is still relevant in the Arctic. With the current trends of thinning sea ice and potentially increasing snowfall rates, snow-ice could become more significant in the Arctic in the future. In another study, we are discussing the snow properties at the sea-ice interface and even in that article, we don't find evidence about snow-ice in the Arctic Ocean:

"*In the manuscript, we briefly mention higher-density snow samples at the sea-ice interface ($\rho >550$ Kg m$^{-3}$) with an enriched $\delta^{18}O$ signal (seen in Figure 3a). We explain that these could be a result of flooding causing a snow-ice layer or a remnant surface scattering layer from the previous melt season. Flooding of the sea ice surface is caused by the weight of the overlying snow submerging the sea ice surface below the ocean surface, causing a negative freeboard. This is typically a process seen in Antarctica due to higher precipitation rates in this region. However, flooding has been recorded close to the marginal ice zone in the Arctic during the 2015 N-ICE cruise, resulting in snow-ice formation (additionally, forcing this is a suggested response to future warming scenarios). This process would introduce a highly saline icy layer at the snow-sea ice interface, with a $\delta^{18}O$ signal of 0 ‰ and a density close to ice $\sim 900$ Kg m$^{-3}$. By investigating the microstructure through micro-CT measurements, and excluding measurements above 550 Kg m$^{-3}$, we find density values in the range typically classified as "snow" ($100 - 550$ Kg m$^{-3}$) with a high isotopic enrichment in the heavy isotopes. As a result, flooding (and a snow-ice layer at the snow-sea ice interface) does not influence our results or discredit the formation of ocean-sourced snow.*" Macfarlane et al. 2023 (https://dx.doi.org/10.21203/rs.3.rs-3572881/v1).*

Also, Itkin et al. (2023) provide valuable insights into the relationship between snow and ice thickness, demonstrating that low snow thicknesses result in no snow-ice occurrence. It is worth mentioning that, as the title suggests, this manuscript is more focused on the snow-atmosphere interface and processes and there is no evidence for collecting snow-ice at this surface layer of snow. We will add a sentence about snow-ice during MOSAiC and cite Macfarlane et al. 2023.

3. I did not see the boxplot for differences in d-excess values in Figure 2. You claimed "Winter months exhibit sharp differences in $\delta^{18}O$ and d-excess values, indicating kinetic fractionation...". On the contrary, perhaps it is the equilibrium fractionation that makes the snow heavier than the water vapor ($>10‰$ for $\delta^{18}O$). In the summer months, the convergence in isotopic signatures between snow and vapor might reflect enriched isotopic signals in snow due to kinetic fractionation. Please check the following refs on the sublimation process in summer:

https://doi.org/10.1029/2018JD030218
https://doi.org/10.5194/tc-2023-76

We show the d-excess in the supplements (Supp Figure 1) but we can also bring that figure together with the current Figure 2 and update Figure 2 if that is the suggestion from the reviewer. Since we mostly focus on $\delta^{18}O$ in our discussion, we initially decided to have the d-excess box plots of surface snow and vapour in the supplements.

I don't agree that in winter the equilibrium fractionation is in play. We have all the conditions (low humidity, negative latent heat fluxes) for sublimation and kinetic fractionation and we see that basically in the negative d-excess values. If the conditions were suitable for equilibrium fractionation, then there was not such a divide in the d-excess values. The reason for observing heavier snow than water vapour ($>10‰$ for $\delta^{18}O$) could be any kind of fractionation.

In summer, the weather conditions in the Arctic are more suitable for equilibrium exchanges between snow and air (higher temperatures, higher humidity, positive latent heat fluxes) and then we see closer values of $\delta^{18}O$ in both snow and vapour contrary to winter.

4. The snow and vapor isotopic signatures are particularly identical during July-Sept. 2020. You could expand on the mechanisms driving this phenomenon besides the vapor deposition, for example, the "supersaturation effect" might occur when the RH is high.

Thanks for the comment. We will consider supersaturation effect as one of the mechanisms driving the similar snow and vapour isotopic signatures.

Deshpande, R.D., Maurya, A.S., Kumar, B., Sarkar, A. and Gupta, S.K., 2013. Kinetic fractionation of water isotopes during liquid condensation under super-saturated condition. Geochimica et Cosmochimica Acta, 100, pp.60-72.

5. For the snow profile, you assumed that the lower layer with enriched $\delta^{18}O$ and lower d-excess values was caused by snow interacting with the sea-ice surface through diffusion of sea-ice. Is it possible that the lower layer of the snow profile is snow-ice?

Not really. We discussed this phenomenon more in detail in Macfarlane et al., 2023. (also see the answer to remark #2.)

6. What are the control factors of the isotopic signals of $\delta^{18}O$ and d-excess? Are the control factors identical or diverse? What kind of changes do the $\delta^{18}O$-$\delta D$ line slopes have in different seasons? Two refs on the interpretation of isotopic signals:

https://doi.org/10.1029/2022JD037037
https://doi.org/10.1029/2018JC013797

The $\delta^{18}O$-$\delta^2H$ comparisons could be misleading here. The whole point of this manuscript is that we don't have the meteoric signal on the surface snow as the snow has been deposited and mostly been affected by post-depositional processes. This is evident in the weak relationship between $\delta^{18}O$ and air temperature and we can observe that in Figure 6 of the manuscript as well as in the previous study (Mellat et al. 2024: https://doi.org/10.1525/elementa.2023.00078).

7. You mentioned the impact of wind-driven snow redistribution. It would be beneficial to include a more detailed analysis of this process and its broader implications for the isotopic homogeneity across the Arctic.

Thanks for the suggestion. A more detailed analysis of impact of wind redistribution metamorphism on snow isotopes requires precise and well-structured laboratory experiments and with the MOSAiC data we are limited in interpreting the impacts of redistribution. Although, we mention this in the text and cite the latest experiment of Wahl et al. on the impacts of wind on snow isotopes changes (https://infoscience.epfl.ch/record/306799?v=pdf).

Although, we have done some analysis on the snow profiles from the ridges with a more homogenous isotopic composition on the upper layer of the ridges compared to the profiles sampled over flat sea ice with increasing trends in $\delta^{18}O$ values from surface to bottom. This points out to the role of wind in picking up the snow deposited on the sea ice in the Arctic and mixing them, and depositing them in the ridges with up to 40 cm of snow with homogenous and depleted $\delta^{18}O$ values.

[Figure]

*Figure 1 Ridges δ¹⁸O profiles.*

We discuss in the last chapter of discussion that the high wind speeds in the Arctic during the expedition contributed to snow redistribution almost at all times. This redistributed snow isotopic signature can be detected in ridges in different times (Figure 1).

8. The authors should consider adding a section on the potential feedback loops that may exist between the observed isotopic changes and larger-scale atmospheric circulation patterns. You already have a discussion using isotope monitoring at Samoylov Island and Ny-Ålesund stations. Please consider using isotopic general circulation model and HYSPLIT trajectory model.

In our manuscript, we concentrate on the isotopic composition analysis of snow deposited on sea ice, which has been in deposited for a few days to weeks. Consequently, the meteoric signal has largely dissipated, meaning that back trajectory analysis is less relevant to learn about the moisture sources.

We have, however, extensively explored this topic in two other publications (Brunello et al., 2023; 2024). In these works, we employed the FLEXPART model to investigate vapour trajectories and their implications for isotopic compositions. Given the detailed examination provided in these papers, we believe it would be redundant to replicate such analysis here.

We appreciate your understanding and hope that our explanation clarifies our focus and methodological choices in this manuscript.

Technical Corrections:

1. In the Abstract, the isotopic terms "$\delta^{18}O$" and "d-excess" is mentioned but not initially defined.

Thanks for the comment. We define these terms in the method.

2. In the whole text, you could use a hyphen in "sea-ice" as an adjective; however, there are many unnecessary hyphens in the words as noun, such as "Arctic sea-ice".

Done.

3. There usually isn't a space before the per mil (‰).

Done.

4. The negative numbers should use the minus symbol instead of hyphen symbol.

Done.

5. Line 20- There is a typo of "d-excess" in the sentence "shows higher $\delta^{18}O$ values and lower d- d-excess values."

Done.

6. Line 105- The "is" should be "are" in the sentence "Sublimation from snow into the atmosphere and deposition from the atmosphere onto the snow is controlled"

Done.

7. In Figure 4, the three boxes overlap together and it's rather hard to distinguish them. I suggest you could use violin plot or grouped boxplot. FYI: https://r-graph-gallery.com/265-grouped-boxplot-with-ggplot2.html

We have considered your suggestion of using grouped boxplots and violin plots. However, implementing grouped boxplots would result in an excessively long figure, given the 12*3 arrangement of boxes side by side. Similarly, while violin plots were also evaluated, they did not provide a visually appealing or effective solution for our dataset.

To address the issue, we have included the precise values in Supplementary Table 1 for readers who need to discern the data more clearly. This supplementary material ensures that all details are accessible and can be interpreted without the visual complexity of the figure.

In "Reference" section, there are a few typos for the isotopic symbols. For example:

Akers, P. D., Kopec, B. G., Mattingly, K. S., Klein, E. S., Causey, D., and Welker, J. M.: Baffin Bay sea ice extent and synoptic moisture transport drive water vapor isotope ( $\delta^{18}$O, $\delta^2$H, and deuterium excess) variability in coastal northwest Greenland, Atmospheric Chemistry and Physics, 20, 13929-13955, 10.5194/acp-20-13929-2020, 2020.

Klein, E. S., Baltensperger, A. P., and Welker, J. M.: Complexity of Arctic Ocean water isotope ($\delta^{18}$O, $\delta^2$H) spatial and temporal patterns revealed with machine learning, Elementa: Science of the Anthropocene, 12, 2024.

Done.

**RC2: 'Comment on egusphere-2024-719', Anonymous Referee #2, 23 Apr 2024**

General comments:

The authors present an exciting dataset from the underexplored central Arctic, a region that plays a key role in the climate system. The authors attempt to characterize the temporal and spatial (or snow type) variability of the isotopic composition of snow on sea ice but are limited by the snow sampling protocol. Nevertheless, some crude patterns are identified, and the dataset is compared to already published datasets of local isotopic water vapor and precipitation isotope data from two remote land-based stations. This comparison is motivated by the identification and deciphering of the drivers forming the snow's isotopic composition. The manuscript is well-written, and the findings are presented in a clean way. With this paper, the authors contribute to the understanding of the polar water cycle by using the isotopic snow signal to identify major surface processes at play. I greatly support this type of study that explores isotope dynamics to improve polar process understanding. However, I am very skeptical about the quality of the presented dataset and thus must raise some major concerns:

The presented isotopic snow dataset is a combination of two publicly available datasets, yet dataset 2 (DS2) has not been peer-reviewed yet. The uncertainty levels given (L. 152) for the DS2 samples measured at the WSL facility are unusually high and increased by a factor of 2 or higher compared to the state- of-the-art (Lis et al., 2008). The resulting uncertainty on d-excess is ~8‰ which makes interpretation difficult. The authors need to justify why they accept such high uncertainty levels.

We had discussions about this correction for many months and the two teams at AWI and SLF decided to choose to correct the dataset with this method after considering many possibilities.

As you can see in the figure below, the main anomaly patterns in each of the datasets in both $\delta^{18}$O and d-excess remain more or less the same and our interpretations of the complete dataset, although we approach them with caution, would be still valid and relatable to the conclusions that we have drawn.

In Figures 2,3,4, we observe the distinct separation between the CiASOM and SLF datasets. The data coverage in the early autumn months is exclusively provided by the CiASOM dataset, as there are no available samples from the SLF dataset during this period. This gap in the SLF dataset limits our ability to analyze isotopic changes comprehensively during early autumn. As winter approaches, we gain access to both datasets, but the characteristics of the data vary significantly. The SLF dataset during winter offers a much higher resolution, though the data tends to be coarser. In contrast, the CiASOM dataset continues to provide information at a similar but lower resolution compared to its autumn data.

Moving into the summer months, specifically from June to September, the SLF dataset becomes much more detailed, with a notably higher resolution of data. This period is critical as it allows for a more granular analysis of isotopic changes in the snow, providing valuable insights into seasonal variations. The CiASOM dataset, however, maintains

a lower resolution throughout this period, underscoring the importance of combining both datasets to achieve a comprehensive view.

To ensure complete coverage of snow isotopes throughout the year, it is essential to integrate both the CiASOM and SLF datasets. Each dataset compensates for the other's limitations, providing a fuller picture of isotopic variations. By examining the ratio of values and the trends of isotope changes, particularly in d18O and d-excess, we observe that the relative changes in values are consistently similar across the two datasets. This consistency allows us to rely on the corrected SLF dataset for discussing the relative changes in isotopes within the snow profiles.

While acknowledging that the data may not be perfectly accurate, it remains sufficiently reliable for analyzing relative changes both within the profiles and across different seasons. The corrected SLF dataset, despite its imperfections, offers a robust framework for understanding seasonal isotopic trends and variations. By leveraging the strengths of both datasets, we can more accurately discuss and interpret the isotopic changes in snow profiles, ensuring a comprehensive and nuanced analysis throughout the year.

[Figure]

*Figure 2 Snow profiles of two datasets.*

*Figure 3 Boxplots of two datasets.*

[Figure]

*Figure 4 distribution of isotopic measurements of CiASOM, SLF, and corrected SLF datastes.*

The authors name evaporation/sublimation fractionation during storage as a reason for the alarmingly high identified bias between the DS2 sub-datasets and apply a linear bias correction. However, as the authors acknowledge themselves in L.80 kinetic fractionation during evaporation greatly influences d- excess. By definition, kinetic fractionation does not affect d18O and dD evenly. Thus, the original d-excess values cannot be "recreated" by applying a linear bias correction which might also explain why some d-excess values (of corrected!) samples are astonishingly low (< - 30‰). Therefore, I have very low confidence in the presented d-excess values and am very skeptical about whether any d-excess interpretation/discussion is meaningful. The author group is aware of this problem as apparent from the

statement in the data publication "We would like to emphasise that calculations of d-excess values for this dataset need interpreting carefully." (Macfarlane et al., 2022), yet the issue is not discussed in this manuscript which is a major flaw.

We acknowledge the complexity introduced by kinetic fractionation, which indeed affects $\delta^{18}O$ and $\delta D$ unevenly and complicates the direct linear correction of d-excess values.

The very negative d-excess values observed in our study are present in both datasets. Similar negative d-excess values have been reported in other Arctic and Antarctic snow-on-sea-ice measurements, particularly at the snow-sea ice interface. This phenomenon is comprehensively discussed in Macfarlane et al. (2023).

We recognize the importance of addressing this issue in our current manuscript. Therefore, we will incorporate a discussion on the implications of kinetic fractionation on d-excess values and emphasize the need for careful interpretation of these values, consistent with our statement in the data publication (Macfarlane et al., 2022). This addition will provide a more nuanced understanding of the challenges and limitations associated with the d-excess data.

L. 163: The bias correction applied to combine the datasets measured in two laboratories is described in one sentence and a reference is given to the online dataset where the same minimal explanation is given. This is clearly not enough to establish trust in the presented dataset correction method. If such extreme correction measures are necessary, the foundation for the correction method should be made accessible somewhere so the reader can judge the quality of the correction method and thus the quality of the data.

Thus, I strongly recommend to either:

- exclude the sub-dataset measured at WSL from the manuscript
- remeasure the samples at a different laboratory (to eliminate the possibility that it is a shortcoming of the laboratory instead of a storage problem)
- at the minimum, indicate clearly which of the presented data points are part of this problematic dataset in the figures and text

Thank you for your detailed feedback regarding the bias correction applied to combine datasets measured in two laboratories. We understand the need for transparency and trust in our dataset correction methods.

Excluding the WSL data from the manuscript is not a feasible option for us, as it would result in the loss of approximately 70% of our data, particularly the critical data from the summer months. Re-measuring the samples in another laboratory would not resolve the issue since the primary cause of the discrepancy has been identified as evaporation after sample collection and during storage. This issue cannot be rectified by additional measurements, regardless of the laboratory used.

To address your concern, we will prepare figures (see answer to reviewer #1) that present the datasets separately in the supplements and ensure that the distinction between them is clearly indicated in both the figures and the text of the manuscript. Additionally, we will provide a more detailed explanation of the bias correction method within the manuscript, allowing readers to better assess the quality of the correction and the data.

The manuscript is rather descriptive and lacks analysis that substantiates the claims being made: correlation is not necessarily equal to causality. Many relevant processes potentially influencing the snow isotope variability are named, but there is a clear need for more elaborate analysis to unambiguously identify responsible drivers including:

- Back trajectory analysis for the comparison to the land-based stations

Thanks for your feedback. We understand the importance of substantiating claims with thorough analysis. However, it is important to note that the snow samples collected during the MOSAiC expedition are not event-based precipitation

samples. As a result, conducting a back-trajectory analysis to determine moisture sources may not provide meaningful insights and could potentially be misleading. The snow collected had been deposited on the sea ice for some time before sample collection, during which it experienced various post-depositional processes. Consequently, the original isotopic composition of the precipitation end-member is unknown, making back-trajectory analysis infeasible for this dataset.

While such analysis can be performed for continental stations, it is not central to the focus of our current manuscript. The primary aim of our study is not to determine the moisture sources for precipitation at Ny-Ålesund and Samoylov Island, but rather to analyze the isotopic composition of snow on sea ice.

- More statistical analysis testing for spatial and temporal trends

Our manuscript includes time series of isotopic compositions and seasonal comparisons. We would appreciate further details on the specific types of temporal analysis the reviewer recommends. The separation of spatial from temporal aspects in the MOSAiC expedition presents a significant challenge, given that Polarstern was continuously drifting. Consequently, our time series provide an integrated spatio-temporal perspective on the data.

If the reviewer is referring to spatial variations within the MOSAiC ice floe, we have addressed this in a separate publication (Mellat et al., 2024), where we present data from different sites on the ice floe and analyze changes in their isotopic compositions.

We believe that our current approach, combining both spatial and temporal aspects, offers a comprehensive view of the data. However, we are open to incorporating additional analyses if the reviewer can specify the desired methods or focus areas.

- The comparison of vapor vs. snow isotope variability by accounting for temperature-dependent equilibrium fractionation to substantiate the claim of non-equilibrium post-depositional processes as drivers for variability

In our previous publication (Mellat et al., 2024), we thoroughly examined the relationship between $\delta^{18}O$ and air temperature. Our findings indicated that the snow on Arctic sea ice is not purely of meteoric origin, which led to the current study and further investigations (Macfarlane et al., 2024). We concluded that non-equilibrium post-depositional processes significantly influence isotope variability. To reinforce our claims in this manuscript, we have included the analysis of the weak seasonal $\delta^{18}O$-temperature correlation in Figure 6.

- A discussion on melt influences on the isotopic composition of the snow

Addressing the influences of melt on isotopic composition requires high-resolution (daily to hourly) measurements of snow isotopes at the same location, along with controlled laboratory experiments. Unfortunately, such high-resolution data were not collected during the MOSAiC expedition.

The primary focus of our manuscript is the interaction between snow and the atmosphere, rather than the effects of melt. Nevertheless, we have included a discussion on melt influences to the extent that our dataset allows. We acknowledge the complexity of this topic and its potential significance, but it falls outside the main scope of our current study.

We appreciate your understanding of these limitations and are open to further discussing melt influences if future high-resolution data become available.

I encourage the authors to reflect on the organization of the manuscript as the discussion section contains several figures presenting relevant data and statistical analysis which should be part of the results. The discussion could then detail the interpretation of these results and draw connections to other literature.

Suggestion: Based on my lack of trust in the combined dataset and the limited statistical analysis and substantiated interpretation I would recommend that the authors consider modifying this manuscript into a dataset paper submission (e.g. Copernicus ESSD journal) which would be a better fit to establish the credibility of the combined dataset, would offer the chance to outline the two land-based stations and the vapor dataset as a possible comparison, and still make this valuable dataset comprehensibly accessible to the community. Alternatively, a clear indication of the problematic data points needs to be made throughout the whole manuscript including the figures or these problematic data need to be excluded from the analyses entirely.

Specific comments:

L. 120: This last statement is not supported by the analysis that follows.

Thanks, we changed the text to:

"*We find that meteoric signals on the snow surface are mostly overprinted by the different post-depositional changes.*"

L. 181: Give specific data uncertainties for all datasets being used

Done.

L. 191: What does "Met City" stand for?

Done.

L. 196: Please indicate somewhere that the observation periods do not overlap

Done.

L. 211: The authors decide to discuss deviations from the median rather than deviations from the (weighted) mean. Please add a short statement of why this approach was chosen. Adding on to this, in the following paragraph "variability" in the dataset is discussed without defining what metric is chosen as the "variability" indicator. Please add.

Done. This sentence has been added:

"*Deviations from the median, rather than the mean, are discussed because the median is less sensitive to extreme values and skewed distributions. This method provides a more robust measure of central tendency for the dataset. By using the median, typical conditions observed throughout the MOSAiC year are more accurately reflected, minimizing the influence of outliers on the interpretation of isotope variations.*"

L. 245: Exchange "typically" with a more specific word

Done.

Fig. 1: Try to find a better way to visualize the "subdivisions" you describe in L. 229 and maybe swap the color coding with the y-axis information to better visualize the variability. Add legend for marker shape.

L. 218: The following paragraph is confusing. It is not clear whether snow type or spatial variability is discussed. Consider adding an extra paragraph for "ridge" isotopes including the definition of ridges

I don't think there is any mention of snow types in this paragraph. It is all about changes of snow height and isotopes. The ridges are already introduced in this paragraph:

*"Ridges, the elevated structures formed on sea ice through the collision and compression of ice floes, serve as deposition zones where re-distributed snow across the Arctic Ocean's sea-ice can gather at greater heights than on the surrounding flat surfaces."*

L. 229: Without establishing the credibility of the d-excess dataset first, the following paragraph is difficult to interpret. Consider marking the problematic data points.

L. 278: Identify possible drivers for the observed "divergence" between vapor and snow by analyzing if vapor equilibrium fractionation can or cannot explain the snow behavior.

Fig 2: How is the precipitation data extracted from ERA5. The respective grid box of the moving ship?

Done.

L. 369: The Macfarlane et al., (2023) manuscript co-authored by the same authors is publicly available on a preprint server but has not been peer-reviewed, so interpretations from this manuscript should be addressed with the necessary precautions.

L. 418: Please elaborate or give details about which processes you expect to be responsible for a "more effective transfer of isotope signals" and provide evidence for the identified processes. An evidence-based credible process identification is missing in section 4.2 in general.

L. 426, L. 608: Please calculate this "disequilibrium" and show the analysis (Aemisegger et al., 2015; Wahl et al., 2024)

Section 4.3: Which snow sample values are being discussed in this section? Does "average daily snow" samples mean that all available samples were averaged? The comparison between snow and vapor isotopic composition (Fig 5b) should account for temperature since equilibrium fractionation is temperature-dependent, even if the effect for d18O is small. So if there is an exchange between snow and vapor, temperature probably has an effect.

L. 438: How was the 2mm aggregated precipitation level chosen? If it's an arbitrary value, how robust is the analysis for different values, e.g. 1.5mm?

Arbitrary value chosen after (Wagner et al., 2023).

Section 4.4: For the comparison with continental locations a back-trajectory analysis seems highly necessary to establish moisture pathways and general synoptic scale conditions.

See answer above to the back-trajectory comment.

L. 545: Exchange "sublimation" with "vapor deposition". A negative latent heat flux is a deposition flux in this dataset.

Done.

L. 549: Commonly, snow sublimation leads to an enrichment in d18O and a decrease in d-excess (Hughes et al., 2021).

Done.

L. 560: The following interpretation of wind redistribution as the driver for surface snow depletion lacks evidence. See also (Wahl et al., 2024)

Fig 7 c: Are the relative humidity levels given referenced against saturation over ice or saturation over water? (Anderson, 1994)

The relative humidity is given against saturation over water.

L. 588: Combined vapor and snow measurements have been performed in the Arctic for at least 10 years (Steen-Larsen et al., 2014), so please rephrase "methodological advancement".

Done.

Technical corrections:

L. 18: Exchange "The Arctic Ocean's snow cover" with "Snow on sea ice"

Done.

L. 24: delete "d-"

Done.

L. 27: «Vapour» or «Vapor»? consistency

Done.

L. 21 vs. L. 71: Consider defining what you mean by "deposition" to avoid confusion. Snowfall deposition or the phase transition between vapor and solid?

Done.

L. 129: Exchange "flow" with "floe"

Done.

Fig 1, caption: Correct plot number labeling

Done.

Fig 1: swap grey and white background coloring

L. 293-303: This is a repetition of information already given in the Data section. Please shorten.

Done.

Fig 4: The line for the median in the Surface Snow MOSAiC Box Plots is hardly visible L. 456: Add respective citation

Done.

Fig 5b: What are empty circles? Indicate the range of precipitation amounts. Have you applied a lower limit to account for the "drizzle effect" in modelled precipitation?

Empty circles for days with less precipitation than 2 mm. No drizzle effect consideration.

L. 535: delete "out"

Done.

L. 598: exchange "snow-sea-ice interactions" with "atmosphere-snow interactions"

Done.

L. 617: Please specify what you mean with this very general statement

Bibliography:

Aemisegger, F., Spiegel, J. K., Pfahl, S., Sodemann, H., Eugster, W., and Wernli, H.: Isotope meteorology of cold front passages: A case study combining observations and modeling, Geophysical Research Letters, 42, 5652–5660, https://doi.org/10.1002/2015GL063988, 2015.

Anderson, P. S.: A method for rescaling humidity sensors at temperatures well below freezing. Journal of Atmospheric and Oceanic Technology, Journal of Atmospheric and Oceanic Technology, 11, 1388+1397, 1994.

Hughes, A. G., Wahl, S., Jones, T. R., Zuhr, A., Hörhold, M., White, J. W. C., and Steen- Larsen, H. C.: The role of sublimation as a driver of climate signals in the water isotope content of surface snow: Laboratory and field experimental results, The Cryosphere, 15, 4949–4974, https://doi.org/10.5194/tc-15-4949-2021, 2021.

Lis, G., Wassenaar, L. I., and Hendry, M. J.: High-Precision Laser Spectroscopy D/H and 18 O/ 16 O Measurements of Microliter Natural Water Samples, Anal. Chem., 80, 287–293, https://doi.org/10.1021/ac701716q, 2008.

Macfarlane, A., Mellat, M., Dadic, R., Meyer, H., Werner, M., Brunello, C., Arndt, S., Krampe, D., and Schneebeli, M.: Ocean-sourced snow: An unaccounted process on Arctic sea ice, https://doi.org/10.21203/rs.3.rs-3572881/v1, 20 November 2023.

Macfarlane, A. R., Schneebeli, M., Dadic, R., Wagner, D. N., Arndt, S., Clemens-Sewall, D., Hämmerle, S., Hannula, H.-R., Jaggi, M., Kolabutin, N., Krampe, D., Lehning, M., Matero, I., Nicolaus, M., Oggier, M., Pirazzini, R., Polashenski, C., Raphael, I., Regnery, J., Shimanchuck, E., Smith, M. M., Tavri, A., Mellat, M., Meyer, H., Werner, M., and Brunello, C. F.: Snowpit stable isotope profiles during the MOSAiC expedition, https://doi.org/10.1594/PANGAEA.952556, 2022.

Steen-Larsen, H. C., Masson-Delmotte, V., Hirabayashi, M., Winkler, R., Satow, K., Prié, F., Bayou, N., Brun, E., Cuffey, K. M., Dahl-Jensen, D., Dumont, M., Guillevic, M., Kipfstuhl, S., Landais, A., Popp, T., Risi, C., Steffen, K., Stenni, B., and Sveinbjörnsdottír, A. E.: What controls the isotopic composition of Greenland surface snow?, Climate of the Past, 10, 377– 392, https://doi.org/10.5194/cp-10-377-2014, 2014.

Wahl, S., Walter, B., Aemisegger, F., Bianchi, L., and Lehning, M.: Identifying airborne snow metamorphism with stable water isotopes, The Cryosphere - Discussions, https://doi.org/10.5194/egusphere-2024-745, 8 April 2024.